

# Nitrogen isotopic composition of plants and soil in an arid mountainous terrain: sunny slope versus shady slope

Chongjuan Chen [1]    Yufu Jia [1]    Yuzhen Chen[1]    Imran Mehmood[1]    Yunting Fang [2]

Guoan Wang [1*]

[1]Beijing Key Laboratory of Farmland Soil Pollution Prevention-control and Remediation, Department of Environmental Sciences and Engineering, College of Resources and Environmental Sciences, China Agricultural University, Beijing, 100193, China.

[2]CAS Key Laboratory of Forest Ecology and Management, Institute of Applied Ecology, Chinese Academy of Sciences, Shenyang 110164, China

*Corresponding author: Guoan Wang

E-mail address: gawang@cau.edu.cn



**Abstract**
Nitrogen cycling is tightly associated with environment. Sunny slope of a given
mountain could significantly differ from shady slope in environment. Thus, N cycling
should also be different between the two slopes. Since leaf $\delta^{15}N$, soil $\delta^{15}N$ and
$\Delta\delta^{15}N_{\text{leaf-soil}}$ ($\Delta\delta^{15}N_{\text{leaf-soil}}$ = leaf $\delta^{15}N$ − soil $\delta^{15}N$) could reflect the N cycling
characteristics, we put forward a hypothesis that leaf $\delta^{15}N$, soil $\delta^{15}N$ and $\Delta\delta^{15}N_{\text{leaf-soil}}$
should differ across the two slopes. However, such a comparative study between two
slopes has never been conducted yet. In addition, environmental effects on leaf and
soil $\delta^{15}N$ derived from studies at global scale were often found to be different from
that at regional scale. This led to our argument that environmental effects on leaf and
soil $\delta^{15}N$ could depend on local environment. To confirm our hypothesis and
argument, we measured leaf and soil $\delta^{15}N$ on the sunny and shady slopes of Mount
Tianshan. Remarkable environment differences between the two slopes provided an
ideal opportunity for our test. The study showed that leaf $\delta^{15}N$, soil $\delta^{15}N$ and
$\Delta\delta^{15}N_{\text{leaf-soil}}$ on the sunny slope were greater than that on the shady slope although the
difference in soil $\delta^{15}N$ was not significant. The result confirmed our hypothesis and
suggested that the sunny slope has higher soil N transformation rates and soil N
availability than the shady slope. Besides, this study observed that the significant
influential factors of leaf $\delta^{15}N$ were temperature, precipitation, leaf N, leaf C/N and
silt/clay ratio on the shady slope, whereas on the sunny slope only leaf C/N was
related to leaf $\delta^{15}N$. The significant influential factors of soil $\delta^{15}N$ were temperature,
precipitation and silt/clay ratio on the shady slope, whereas on the sunny slope MAP



and soil moisture exerted significant effects. Precipitation exerted contrary effects on
soil $\delta^{15}N$ between the two slopes. Thus, this study supported our argument that the
relationships    between    leaf    and    soil    $\delta^{15}N$    and    environmental    factors    are
local-dependent.

**1 Introduction**
In natural terrestrial ecosystem, nitrogen (N) is not only the most required element,
but also is usually a key limiting resource for plants (Vitousek et al., 1997), thus,
studying N cycling is of vital importance. The variations of nitrogen isotope ratio
($\delta^{15}N$) in plants and soil are tightly associated with many biogeochemical processes
including    N    mineralization,    ammonia    volatilization,    nitrification,    denitrification
(Högberg, 1997; Houlton et al., 2006). Mineralization produces available N, including
ammonium    and    nitrate,    which    are    the    substrates    for    ammonia    volatilization,
nitrification and denitrification. During these processes, gaseous N loss is more likely
to be depleted in $^{15}N$, which will cause the remaining N pool and subsequently plants
to enrich $^{15}N$ (Högberg, 1997). Additionally, the difference between leaf $\delta^{15}N$ and soil
$\delta^{15}N$ ($\Delta\delta^{15}N_{leaf\text{-}soil}$ = leaf $\delta^{15}N$ – soil $\delta^{15}N$), which is also named enrichment factor
(Emmett et al., 1998), was also suggested to be an indicator of ecosystem N cycling
(Charles and Garten, 1993; Kahmen et al., 2008; Fang et al., 2011), and it was also
reported    to    be    correlated    with    soil    N    transformation    rates    (N    mineralization    or
nitrification rates) (Garten and Van Miegroet, 1994). Thus, nitrogen isotopes have
been widely applied in studies of terrestrial ecosystem N cycling (Handley et al., 1999;





Evans, 2001; Robinson, 2001; Hobbie and Colpaert, 2003; Houlton et al., 2007).

For a given mountain, its sunny slope may be significantly different from its shady

slope in climate and environment. It is well known that ecosystem N cycling is
associated with climatic and environmental conditions (Amundson et al., 2003; Craine
et al., 2009; Yang et al., 2013; Zhou et al., 2016), thus, ecosystem N cycling should
vary across sunny and shady slopes. Since leaf $\delta^{15}N$, soil $\delta^{15}N$ and $\Delta\delta^{15}N_{leaf-soil}$ could
reflect and indicate ecosystem N cycling, differences in leaf $\delta^{15}N$, soil $\delta^{15}N$ and
$\Delta\delta^{15}N_{leaf-soil}$ were expected to appear between sunny and shady slopes. Comparisons
on leaf $\delta^{15}N$, soil $\delta^{15}N$ and $\Delta\delta^{15}N_{leaf-soil}$ across two slopes of a mountain would
provide a good insight into the response of terrestrial ecosystem N cycling to climate
and environment. However, to our knowledge, such a comparative study has never
been conducted yet.

Most of the published works consistently suggested that leaf $\delta^{15}N$ increased with

increasing mean annual temperature (MAT) and decreasing mean annual precipitation
(MAP) at large regional or global scales (Austin and Sala, 1999; Amundson et al.,
2003; Craine et al., 2009). However, in contrast to the commonly reported patterns,
leaf $\delta^{15}N$ was found to be negatively related to MAT in some Asian regions, e.g., in
Inner Mongolian (Cheng et al., 2009) and eastern China (Sheng et al., 2014). Relative
to plant $\delta^{15}N$, soil $\delta^{15}N$ has been little addressed. Some studies demonstrated that soil
$\delta^{15}N$ decreased with increasing MAP and decreasing MAT at the global scale
(Amundson et al., 2003; Craine et al., 2015b). However, studies based on local or
region scale showed inconsistent results with the global patterns. Cheng et al. (2009)





reported that soil $\delta^{15}N$ increased with decreasing MAT in Inner Mongolian. Sheng et
al. (2014) showed that the soil $\delta^{15}N$ in tropical forest ecosystems were $^{15}N$-depleted
than in temperate forest ecosystems of eastern China. Yang et al. (2013) found that
soil $\delta^{15}N$ did not vary with either MAT or MAP on the Tibetan Plateau. Wang et al.
(2014) revealed a second-order polynomial relationship between soil $\delta^{15}N$ and aridity
index across arid and semi-arid regions. The above inconsistent observations led to
our argument that the relationships between environmental factors and leaf $\delta^{15}N$ or
soil $\delta^{15}N$ would depend on local environment. Comparisons on the effects of climatic
and environmental factors on leaf $\delta^{15}N$ and soil $\delta^{15}N$ between sunny and shady slopes
of a given mountain could test the argument.

This study was conducted on the sunny slope and shady slope of Mount Tianshan.

It is an ideal place for testing our hypotheses because its sunny slope differs greatly
from its shady slope in climatic and environmental conditions (Deng et al., 2015;
Zhang et al., 2016). The first objective of the present study was to confirm our
hypothesis that the sunny slope differs from the shady slope in leaf $\delta^{15}N$, soil $\delta^{15}N$ and
$\Delta\delta^{15}N_{leaf-soil}$. The second objective was to test our argument that environmental effects
on leaf $\delta^{15}N$ and soil $\delta^{15}N$ are local-dependent.

## 2 Materials and methods

### 2.1 Study area

Mount Tianshan is one of the largest seven mountains over the world. It has a total
length of 2500 km straddling four countries including China, Kazakhstan, Kyrgyzstan



and Uzbekistan. In China, Mount Tianshan stretches 1700 km along the east-west
direction in the Xinjiang Uygur Autonomous Region and covers about 570,000 square
kilometers and accounts for one third of the whole area. Mount Tianshan divides
Xinjiang into two parts, the south of Tianshan is the Tarim Basin and the north is the
Dzungaria Basin.
This study was conducted along an elevation transect on the shady and sunny
slopes on eastern Mount Tianshan (42.43 ° – 43.53 °N, 86.23 ° – 87.32 °E) (Fig.1).
Mount Tianshan is characterized by an arid mountainous climate; vertical variations
in temperature and precipitation are very pronounced, temperature decreases and
precipitation increases with altitude on both slopes. The shady slope differs
significantly from the sunny slope both in climate and vegetation. On the shady slope,
the annual mean temperature (MAT) ranges from -6.40 ℃ to 3.90 ℃ with the
average temperature of -1.85 ℃, and the annual mean precipitation (MAP) ranges
from 314 mm to 472 mm with the average precipitation of 402 mm. While on the
sunny slope, the MAT varies from -5.65 ℃ to 9.23 ℃ with the average temperature
of 1.03 ℃, and the MAP varies from 124 mm to 308 mm with the average
precipitation of 246 mm. There were four meteorological observatories along our
elevation transects, two on either slope of Mount Tianshan (Table 1).
Fig. 1
Table 1



Intact and continuous vertical vegetation and soil spectrums can be observed along
the two slopes. On the shady slope from bottom to top, vegetation spectrum consists
of upland desert (800–1100 m), upland steppe (1100–2500 m), frigid coniferous forest
(1800–2700 m), subalpine meadow (2500–3300 m), alpine meadow (3000–3700 m),
alpine sparse vegetation and a desert zone (3700–3900 m), and an alpine
ice-and-snow zone (> 3900 m). A corresponding soil spectrum on shady slope
includes brown calcic soil (800–1100 m), chestnut soil (1100–2500 m), mountain
grey cinnamon forest soil (1800–2700 m), subalpine meadow soil (2500–3300 m),
alpine meadow soil (3000–3700 m) and chilly desert soil (> 3700 m). While on the
sunny slope, it includes upland desert (1300–1800 m), arid upland steppe (1800–2600
m), subalpine steppe (2600–2800 m), alpine meadow and cushion plants (2800–3800
m), an alpine desert zone (3800–4000 m), and an alpine ice-and-snow zone (> 4000
m). The corresponding soil spectrum of sunny slope consists of sierozem (1300–1800
m), chestnut soil (1800–2800 m), alpine meadow soil (2800–3800 m), and chilly
desert soil (> 3800 m).

**2.2 Plants and soil sampling**
An altitudinal transect of 1,564 to 3,800 m above sea level (a.s.l.) was set on the
shady slope, and 1,300 to 3,780 m a.s.l. on the sunny slope. Few human habitats
distribute along the two transects. Plant and soil samples were collected in July of
2014. To minimize the influences of human activities, light regime, or location within
the canopy, the sampling was restricted to open sites that are far from the major roads



and human habitats.
Plants and soil were collected along the two transects at altitudinal intervals of
about 100 m. Almost all plant species that we found at each sampling site were
collected, and at each site, 5–7 individual plants of each species were collected and
the same number of leaves was sampled from each individual plant. For shrubs and
herbs, the uppermost leaves of each individual plant were collected; for tree species, 8
leaves were collected from each individual, 2 leaves were collected at each of the 4
cardinal directions from the positions of full irradiance, about 8–10 m above the
ground. The leaves from the same species of each site were combined into one sample.
Excluding N-fixing plants and mosses, a total of 90 plant samples were collected from
shady slope, including 72 herbs and 18 woody plants; 105 on the sunny slope,
covering 85 herbs and 20 woody plants.
Surface soils (0–5cm) were collected after removing the litter layer at each
sampling site. At each location, one composite soil sample was prepared by
combining six subsamples randomly taken within a radius of 20 m. Sample was used
to determine soil index including $\delta^{15}N$, N content, silt/clay ratio, pH and particle size.
In addition, at each sampling site, we also collected another three soil samples using a
ring, which were used to measure soil bulk density and moisture.

**2.3 Laboratory measurements**
Plant and soil samples were air-dried in the field and then in the laboratory. The soil
samples were sieved through a 2 mm sieve to remove stones and plant residues. Plant





leaves and about 5 g sieved soil samples were then ground into a fine powder using a
steel ball mixer mill MM200 (Retsch GmbH, Haan, Germany). $\delta^{15}N$, N and C
contents in leaves, and $\delta^{15}N$ and N contents in soil were measured on a Delta$^{Plus}$ XP
mass spectrometer (Thermo Scientific, Bremen, Germany) coupled with an automated
elemental analyzer (Flash EA1112, CE Instruments, Wigan, UK) in a continuous flow
mode at the Stable Isotope Laboratory of the College of Resources and Environmental
Sciences, China Agricultural University. For this measurement, we obtained standard
deviations of less than 0.1% for C and N contents and less than 0.15‰ for $\delta^{15}N$
among replicates of the same sample.
The measurements of soil pH and soil particle size (clay, silt and sand content)
were determined using the sieved soil samples. Soil pH was measured using the pH
electrode in soil water suspension, with soil to water ratio of 1:2.5 (10 g soil and 25
mL deionized water removing carbon dioxide). Soil particle size (clay, silt and sand
content) was analyzed using a particle size analyzer (Malvern Masterizer 2000, UK)
after removing the calcium carbonates and organic matter. Soil moisture and bulk
density were determined after oven drying at 105 ± 2 ℃ to a constant weight. Soil
moisture of each sample was the difference between its wet and dry weight divided by
its dry weight. Soil bulk density was the dry weight divided by the certain volume of
the ring.

**2.4 Statistical analysis**
The MAT and MAP data of each sampling elevation used in the statistical analyses





were generated by interpolation based on the climatic data derived from the four
meteorological observatories distributed along the altitudinal transect. Statistical
analyses were conducted by SPSS software (SPSS for Windows, Version 20.0,
Chicago, IL, USA). One-way analysis variance (ANOVA) was used to compare leaf
$\delta^{15}N$, soil $\delta^{15}N$ and $\Delta\delta^{15}N_{leaf-soil}$ between the shady and sunny slopes. Leaf C/N was
ln- transformed to improve data normality. The relationships between $\Delta\delta^{15}N_{leaf-soil}$ and
leaf $\delta^{15}N$ were performed by the linear regression on the two slopes. Leaf and soil
$\delta^{15}N$ were firstly analyzed by multiple linear regressions against all potential
influential factors using ordinary least square (OLS) estimation. The potential
influential factors of leaf $\delta^{15}N$ included MAP, MAT, leaf N content, leaf C/N, soil
$\delta^{15}N$, soil N content, silt/clay ratio, soil moisture, soil bulk density and soil pH. The
potential influential factors of soil $\delta^{15}N$ consisted of MAP, MAT, soil N content,
silt/clay ratio, soil moisture and soil bulk density and soil pH. Finally, correlation
analyses were conducted to explore the effects of these factors on leaf $\delta^{15}N$ and soil
$\delta^{15}N$.

### 3 Results

**3.1 Comparisons of $\delta^{15}N$ in leaf and soil between the shady and the sunny slopes**

On Mount Tianshan, for all species pooled together, the arithmetic mean (mean $\pm$ SE)
of leaf $\delta^{15}N$ were 0.5 ± 0.2‰ and 2.0 ± 0.2‰ for the plants grown on the shady and
the sunny slopes, respectively. One-way ANOVA suggested a significant difference
for leaf $\delta^{15}N$ between the shady and sunny slopes ($P < 0.001$) (Fig. 2a). The mean soil



$\delta^{15}N$ of the shady and sunny slope were 4.1 ± 0.4‰ and 5.0 ± 0.8‰, respectively.
One-way ANOVA showed that sampling slope exerted no significant effect on soil
$\delta^{15}N$ ($P = 0.290$) (Fig. 2b). The mean $\Delta\delta^{15}N_{leaf-soil}$ was -3.6 ± 0.3‰ for the shady
slope and -2.4 ± 0.3‰ for the sunny slope, and one-way ANOVA suggested a
significant difference in $\Delta\delta^{15}N_{leaf-soil}$ between the two slopes ($P = 0.003$) (Fig. 2c). In
addition, this study showed that $\Delta\delta^{15}N_{leaf-soil}$ was positively related to $\delta^{15}N_{leaf}$ on the
two slopes ($P < 0.001$, Fig. 3).
Fig. 2
Fig. 3

**3.2 The relationships between leaf $\delta^{15}N$ and potential influential factors**
A multiple regression of leaf $\delta^{15}N$ against potential influential factors including soil
$\delta^{15}N$, MAT, MAP, leaf N content, leaf C/N, soil N content, soil moisture, soil pH, soil
bulk density and silt/clay was conducted. The statistical analyses showed that 45.8%
and 23.4% of the variability in leaf $\delta^{15}N$ on the shady slope and sunny slope could be
explained as a linear combination of all 10 independent variables, respectively ($P <$
0.001 for the shady slope and $P = 0.005$ for the sunny slope) (Table 2). Among these
influential factors, MAT, leaf N content correlated positively and MAP, leaf C/N, and
silt/clay ratio correlated negatively with leaf $\delta^{15}N$ on the shady slope (Table 3).
Whereas on the sunny slope, only leaf C/N was found to have a negative effect on leaf
$\delta^{15}N$, MAP correlated marginally and negatively with leaf $\delta^{15}N$ (Table 4).
Table 2



Table 3
Table 4

**3.3 The relationships between soil δ$^{15}$N and potential influential factors**

Multiple regressions analysis with soil δ$^{15}$N as a dependent variable and MAT, MAP,
soil N content, silt/clay ratio, soil moisture, soil bulk density and soil pH as
independent variables were conducted separately for the shady slope and the sunny
slope. The statistical analyses showed that the regressions were very significant on
both slopes ($P < 0.001$ for the both slopes). The seven factors in total accounted for
55.2% and 72.7% of soil δ$^{15}$N variance on the shady and sunny slope, respectively
(Table 2). Considering the potential link between soil N and plant N, new multiple
regressions including leaf δ$^{15}$N, leaf N and leaf C/N were performed on the two slopes.
Compared to the old multiple regressions, the new regressions did not exhibit changes
in $R^2$ and $P$ values on both slopes (in the new regressions $P < 0.001$ and $R^2 = 0.563$
for the shady slope, $P < 0.001$ and $R^2 = 0.738$ for the sunny slope). Furthermore,
compared to the adjusted $R^2$ values derived from the old regressions (adjusted $R^2 =$
0.513 for the shady slope, adjusted $R^2 = 0.708$ for the sunny slope), the values of the
new regressions were smaller or almost unchanged (adjusted $R^2 = 0.506$ for the shady
slope, adjusted $R^2 = 0.709$ for the sunny slope) (Table 2). Thus, the new multiple
regressions indicated no effect of leaf nutrient traits on soil δ$^{15}$N. Among these factors,
only MAT, MAP and silt/clay were found to be significantly related to the soil δ$^{15}$N of
the shady slope. The soil δ$^{15}$N was observed to increase with increasing MAP and



decreasing MAT and silt/clay ratio on the shady slope (Table 3). However, on the
sunny slope, only MAP and soil moisture were found to play a significant and
negative role in soil $\delta^{15}N$ (Table 4).

**4 Discussion**
**4.1 Differences in leaf $\delta^{15}N$, soil $\delta^{15}N$ and $\Delta\delta^{15}N_{leaf\text{-}soil}$ between the sunny and the**
**shady slopes**
On Mount Tianshan, leaf $\delta^{15}N$ and $\Delta\delta^{15}N_{leaf\text{-}soil}$ both showed higher values on the
sunny slope than the shady slope; soil $\delta^{15}N$ of the sunny slope was also more positive
than that of the shady slope although the difference was not significant. The results
confirmed our hypothesis that, for a given mountain, the leaf $\delta^{15}N$, soil $\delta^{15}N$ and
$\Delta\delta^{15}N_{leaf\text{-}soil}$ of the sunny slope could differ from those of the shady slope. Greater leaf
$\delta^{15}N$ on the sunny slope than shady slope suggested that the sunny slope had higher
soil N availability and higher soil N transformation rates (N mineralization or
nitrification rates) (Garten and Van Miegroet, 1994; McLauchlan et al., 2007).
Increasing soil N transformation rates led to an increase in soil available N.
Meanwhile, increasing soil N transformation rates could result in more $^{15}N$
enrichment in soil available N sources, and consequently plant $\delta^{15}N$ increased,
because N transformation processes discriminate against $^{14}N$.
$\Delta\delta^{15}N_{leaf\text{-}soil}$ was greater on the sunny slope than the shady slope. This result also
suggested that the sunny slope has higher N availability and N mineralization or
nitrification rates relative to the shady slope, because previous studies reported that





$\Delta\delta^{15}N_{leaf-soil}$ increased with increasing soil N transformation rates (N mineralization or
nitrification rates) and N availability (Garten and Van Miegroet, 1994; Kahmen et al.,
2008; Cheng et al., 2010).
In most natural ecosystems, where soil available N is limited or plant N demand
exceeds soil nitrogen supply, N isotopic discrimination would be negligible during
nitrogen uptake and assimilation (Högeberg et al., 1999). Thus, leaf $\delta^{15}N$ is a good
approximation to $\delta^{15}N$ of soil available nitrogen sources in natural ecosystems
(Virginia and Delwiche, 1982; Cheng et al., 2010; Craine et al., 2015a), and
$\Delta\delta^{15}N_{leaf-soil}$ was interpreted as the isotopic composition of plant-available N
(Amundson et al., 2003). So, both leaf $\delta^{15}N$ and $\Delta\delta^{15}N_{leaf-soil}$ are good indicators of
available N, and the positive relationship is expected to exist between leaf $\delta^{15}N$ and
$\Delta\delta^{15}N_{leaf-soil}$. The results that $\Delta\delta^{15}N_{leaf-soil}$ was highly correlated with leaf $\delta^{15}N$ on the
two slopes of Mount Tianshan provided a powerful support for the above viewpoints.
The observed positive relationship between $\Delta\delta^{15}N_{leaf-soil}$ and leaf $\delta^{15}N$ suggested that
significant difference in $\Delta\delta^{15}N_{leaf-soil}$ between the sunny and shady slopes was mainly
due to the significant difference in leaf $\delta^{15}N$ between two slopes.

**4.2 Influences of varied factors on leaf $\delta^{15}N$ and soil $\delta^{15}N$: sunny slope versus**
**shady slope**
The regression and correlation analyses showed that each factor did not exert
completely identical effect on leaf $\delta^{15}N$ and soil $\delta^{15}N$ across the two slopes, this
provided powerful support for our argument that the influences of environmental





factors on leaf $\delta^{15}$N and soil $\delta^{15}$N are local-dependent.
Leaf C/N may play a role in regulating biogeochemical cycles of carbon and
nitrogen in natural ecosystems (Luo et al., 2004), or, conversely, soil biogeochemistry
and plant physiology also cause the shifts in leaf C/N stoichiometric characters (Reich
and Oleksyn, 2004; Yang et al., 2011). In this study, leaf C/N was negatively
correlated with leaf $\delta^{15}$N on both slopes of Mount Tianshan. The result was similar to
the finding by Pardo et al. (2006), in which leaf $\delta^{15}$N and root $\delta^{15}$N both decreased
with forest floor C/N. A negative correlation between leaf C/N and $\delta^{15}$N was also
reported for the fine roots in Glacier Bay (Hobbie et al., 2000). Two possible reasons
were responsible for the pattern observed in the present study. First, the increase in
leaf C/N might be caused by enhanced photosynthesis, which would aggravate the
limit in nitrogen nutrients and result in a decrease in nitrogen availability,
consequently, plants would deplete $^{15}$N. Second, leaf C/N usually was considered to
be negatively correlated with leaf N contents because leaf C contents always keep
relative stable (Tan and Wang, 2016). The relative stability of leaf C was also
observed in this study. The negative relationship between leaf C/N and leaf $\delta^{15}$N
might be caused by the positive relationship between leaf N content and leaf $\delta^{15}$N,
which has been reported by many studies (Chen et al., 2015; Zhang et al., 2015;
Craine et al., 2012; Pardo et al., 2006; Craine et al., 2009; Martinelli et al., 1999).
This study also found a positive relationship between leaf N content and leaf $\delta^{15}$N on
the shady slope of Mount Tianshan.
MAP was observed to be significantly and negatively correlated with leaf $\delta^{15}$N on



the shady slope; however, on the sunny slope the relationship was just marginally
significant. A negative relationship between leaf $\delta^{15}N$ and MAP was reported in many
previous studies (Austin and Sala, 1999; Handley et al., 1999; Robinson, 2001;
Amundson et al., 2003; Craine et al., 2009). The decrease in leaf $\delta^{15}N$ with increasing
precipitation could be associated with decreased gaseous N loss in wetter regions
(Houlton et al., 2006).

MAT played a positive effect in the leaf $\delta^{15}N$ of the shady slope, whereas on the

sunny slope the effect of MAT was not observed. The probable explanation for this
observation was that the climate on the shady slope is very cold (the average MAT =
-1.85 ℃), temperature is the key growth-limiting factor for plants, thus, temperature
exerted an effect on leaf $\delta^{15}N$. However, on the sunny slope the climate is relatively
warm except those sites with higher altitudes, and usually, temperature does not limit
plant growth, thus, leaf $\delta^{15}N$ was not related to temperature.

On Mount Tianshan, soil $\delta^{15}N$ increased with increasing MAP on the shady slope,

while decreased with increasing MAP on the sunny slope. Soil $\delta^{15}N$ could be
determined by the balance of the N input or output processes and corresponding
isotopic fractionation factors (Brenner et al., 2001; Bai and Houlton, 2009; Wang et
al., 2014). Considering that the leaching loss could be neglected on both slopes
because of the dry environment, soil $\delta^{15}N$ can be estimated by the following equation:

Soil $\delta^{15}N = \delta^{15}N_{input} + \varepsilon_G \times f_G + \varepsilon_P \times f_P$                (1)

where $\delta^{15}N_{input}$ is the input $\delta^{15}N$; $f_G$ and $f_P$ are the fraction of gas losses and net plant
N accumulation out of total N losses (%), respectively; $\varepsilon_G$ and $\varepsilon_P$ are the fractionation



factors of corresponding N losses processes, respectively. And
$$f_G + f_P = 1 \qquad (2)$$
$\varepsilon_G$ varies from 16‰ to 30‰ (Handley et al., 1999; Robinson, 2001); $\varepsilon_P$ is between 5‰
and 10‰ (Handley et al., 1999; Evans, 2001), thus, in general, $\varepsilon_G > \varepsilon_P$, and soil $\delta^{15}N$
is correlated positively with $f_G$ and negatively with $f_P$ based on eqn. (1) and (2). On
the shady slope, rainfall event may accelerate the gas losses (nitrification and
denitrification processes) more than plant N uptake, while it may be opposite on the
sunny slope. On the shady slope, with increase in MAP, $f_G$ increases and causes $^{15}N$
enrichment in soil; on the sunny slope, $f_P$ increases with MAP, and results in $^{15}N$
depletion in soil.
The effects of silt/clay ratio on soil $\delta^{15}N$ might be driven by the indirect effects of
silt/clay ratio on soil moisture and soil oxygen concentrations. The shady slope is
wetter than the sunny slope, and the shady slope will prefer denitrification, while
nitrification will be favored on the sunny slope. On the shady slope, with increase in
silt/clay ratio, soil oxygen concentration increases and this inhibits soil denitrification,
consequently, $^{15}N$ depletion in soil would be resulted in, thus silt/clay ratio showed a
negative relationship with soil $\delta^{15}N$.

**5 Conclusion**
We sampled plants and soils on the sunny slope and shady slope of Mount Tianshan
and measured their $\delta^{15}N$. Sunny slope differs significantly in climate and environment
from shady slope. In the present study, leaf $\delta^{15}N$ and $\Delta\delta^{15}N_{leaf-soil}$ ($\delta^{15}N_{leaf} - \delta^{15}N_{soil}$)

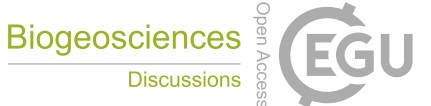



of the sunny slope were more positive than that of the shady slope, soil $\delta^{15}N$ of the
sunny slope was also higher than that of the shady slope although the difference
between the two slopes was not significant. The results suggested that the sunny slope
has higher soil N transformation rates and soil N availability relative to the shady
slope. In addition, among the potential influential factors, MAP, leaf C/N and silt/clay
ratio had negative effects while MAT and leaf N content had positive effects on leaf
$\delta^{15}N$ of the shady slope; however, on the sunny slope, only leaf C/N played a negative
role in leaf $\delta^{15}N$. For soil $\delta^{15}N$, the significant influential factors were MAT, MAP and
silt/clay ratio on the shady slope, whereas on the sunny slope, MAP and soil moisture
exerted significant effects. Interestingly, MAP was found to exert contrary effects on
soil $\delta^{15}N$ between the two slopes. This indicated that environmental influences on leaf
$\delta^{15}N$ and soil $\delta^{15}N$ are local-dependent.

*Data availability*. There is no underlying material and related items in this paper. All
data will be provided in the Supplement.

*Competing financial interests*. The authors declare no competing financial interests.

*Acknowledgments*. This research was supported by the Chinese National Basic
Research Program (No. 2014CB954202) and the National Natural Science Foundation
of China (No. 41272193). We would like to thank Ma Yan for analyzing nitrogen
isotopes at the Stable Isotope Laboratory of the College of Resources and



Environment, China Agricultural University.

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





# Figures


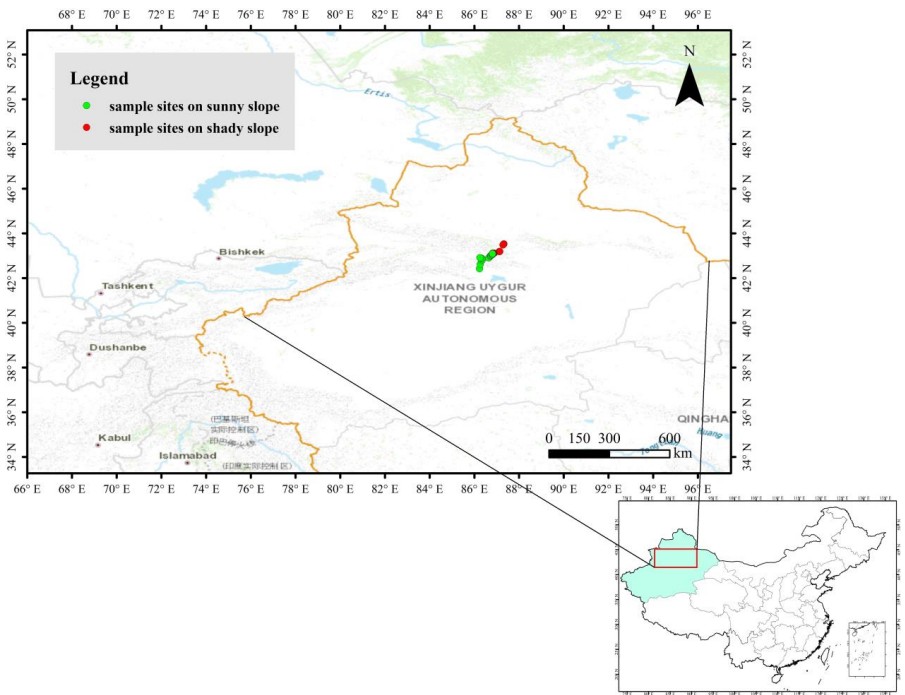


**Fig 1. Sketch of study area. Locations of the sampling sites are indicated with points. A total**

**of 17 sites (red dots) were selected on the shady slope, and 16 sites (green dots) on the sunny**

**slope.**












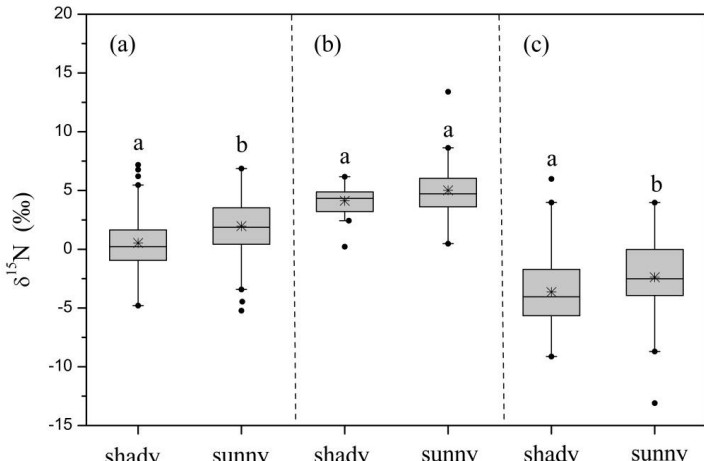


**Fig 2. Differences in (a) leaf $\delta^{15}N$, (b) soil $\delta^{15}N$ and (c) $\Delta\delta^{15}N_{leaf\text{-}soil}$ between the shady and**

**sunny slopes of Mount Tianshan.** Each box represents range of middle 50% of group values, the

center lines and points within the boxes are median and mean values. Whiskers are outside 25%

each, and dots are outliers.















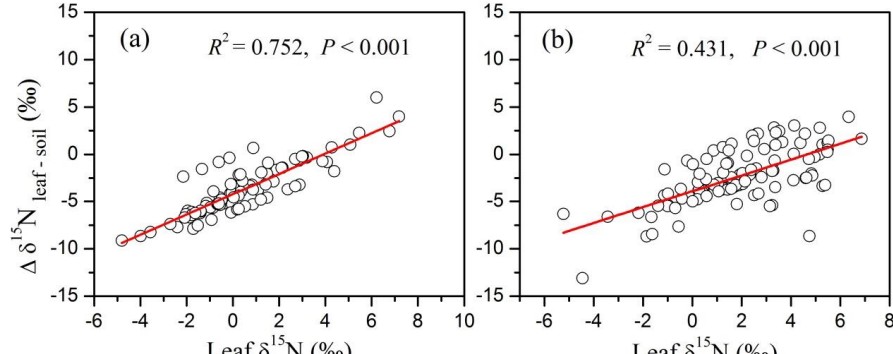


**Fig 3. Relationships between $\Delta\delta^{15}N_{leaf\text{-}soil}$ and leaf $\delta^{15}N$ on the shady slope (a) and the sunny**

**slope (b) of Mount Tianshan.**



















# Tables


**Table 1. Climate data from the meteorological observatories in the research area**

| Meteorological observatories | Locations | MAT/℃ | MAP/mm | Alt./m |
|---|---|---|---|---|
| WLMQ | shady slope | 6.9 | 269.4 | 918.7 |
| MOS | shady slope | -5.2 | 453.4 | 3539.0 |
| BLT | sunny slope | 6.6 | 208.4 | 1738.3 |
| YQ | sunny slope | 8.4 | 73.3 | 1055.8 |

*Abbreviation: WLMQ, Wulumuqi Meteorological Observatory; MOS, Mountain Observation*
*Station of the Tianshan Glaciological Station, Chinese Academy of Sciences; BLT, Baluntai*
*Meteorological Observatory; YQ, Yanqi Meteorological Observatory; MAT, annual mean*
*temperature; MAP, annual mean precipitation.*












**Table 2. Multiple linear regressions of leaf $\delta^{15}$N and soil $\delta^{15}$N based on ordinary least square**
**(OLS) estimation.**

| Model | Dependent variable | Shady slope | | | Sunny slope | | |
|---|---|---|---|---|---|---|---|
| | | $R^2$ | Adjusted $R^2$ | $P$ | $R^2$ | Adjusted $R^2$ | $P$ |
| 1 | Leaf $\delta^{15}$N | 0.458 | 0.388 | < 0.001 | 0.234 | 0.150 | 0.005 |
| 2 | Soil $\delta^{15}$N | 0.552 | 0.513 | < 0.001 | 0.727 | 0.708 | < 0.001 |
| 3 | Soil $\delta^{15}$N | 0.563 | 0.506 | < 0.001 | 0.738 | 0.709 | < 0.001 |

*Note: In the model-1, independent variables were MAT, MAP, leaf N content, leaf C/N, soil $\delta^{15}$N,*
*soil N content, silt/clay, soil moisture, soil bulk density and soil pH. In the model-2, independent*
*variables were MAT, MAP, soil N content, silt/clay, soil moisture, soil bulk density and soil pH. In*
*the model-3, besides all variables in Model-2, leaf $\delta^{15}$N, leaf N content and leaf C/N were also*
*included in independent variables.*















**Table 3. Correlation analyses between leaf or soil $\delta^{15}$N and influential factors on the shady**
**slope of Mount Tianshan.**

| | Leaf $\delta^{15}$N | | Soil $\delta^{15}$N | |
|---|---|---|---|---|
| | *r* | *P* | *r* | *P* |
| Leaf $\delta^{15}$N | 1 | --- | -0.120 | 0.264 |
| Soil $\delta^{15}$N | -0.120 | 0.264 | 1 | --- |
| MAT | **0.266** | 0.012 | **-0.385** | < 0.001 |
| MAP | **-0.272** | 0.010 | **0.387** | < 0.001 |
| Leaf N content | **0.340** | 0.001 | -0.090 | 0.397 |
| Leaf C/N | **-0.452** | < 0.001 | -0.036 | 0.739 |
| Soil N content | -0.048 | 0.659 | 0.088 | 0.408 |
| Soil moisture | 0.005 | 0.962 | -0.061 | 0.565 |
| Soil pH | 0.162 | 0.132 | 0.070 | 0.513 |
| Soil bulk density | -0.056 | 0.604 | 0.145 | 0.174 |
| Silt/clay ratio | **-0.236** | 0.027 | **-0.370** | < 0.001 |

*Note: the r values were in bold when P < 0.05.*














**Table 4. Correlation analyses between leaf or soil $\delta^{15}N$ and influential factors on the sunny**

**slope of Mount Tianshan.**

|  | Leaf $\delta^{15}N$ | | Soil $\delta^{15}N$ | |
|---|---|---|---|---|
|  | r | P | r | P |
| Leaf $\delta^{15}N$ | 1 | --- | 0.175 | 0.074 |
| Soil $\delta^{15}N$ | 0.175 | 0.074 | 1 | --- |
| MAT | 0.157 | 0.109 | 0.115 | 0.244 |
| MAP | -0.168 | 0.087 | **-0.203** | 0.038 |
| Leaf N content | 0.119 | 0.229 | -0.073 | 0.459 |
| Leaf C/N | **-0.228** | 0.021 | 0.062 | 0.533 |
| Soil N content | -0.173 | 0.078 | 0.014 | 0.888 |
| Soil moisture | -0.141 | 0.150 | **-0.229** | 0.019 |
| Soil pH | 0.04 | 0.686 | -0.138 | 0.161 |
| Soil bulk density | 0.151 | 0.125 | 0.041 | 0.679 |
| Silt/clay ratio | -0.07 | 0.477 | -0.004 | 0.964 |

*Note: the r values were in bold when P < 0.05.*





