# Peer review of "Nitrogen isotopic composition of plants and soil in an arid mountainous terrain: south slope versus north slope Chongjuan Chen 1 Yufu Jia 1 Yuzhen Chen 1 Imran Mehmood 2 Yunting Fang 3 Guoan Wang 1\* 1Beijing Key Laboratory of Farmlan"

_Biogeosciences, 2017_

## Referee Comment (RC1) · Anonymous Referee #1 · 3 Oct 2017

The authors compared the leaf and soil 15N isotopic labels on two sides of mountain where one slope is sunny and the other one is shady. They hypothesized that due to different environmental conditions, both slopes should differ in terms of these signatures. Moreover, they hypothesized that the local environmental conditions determine these variables instead of global generalization of environmental factors. The authors did extensive sampling of leaves and soils across a gradient of > 2000 m a.s.l. on both slopes and measured the $\delta$15N of these samples. The introduction and materials and methods are well written. The description of results is also adequate, although it can be improved in the light of the suggestion given below. However, the discussion in it's current form is patchy and lacks the insight with which the results could have been

explained. Briefly, the reasoning behind different results and the varying environmental factors determining them on two slopes is not clear. In addition, the correlation of various environmental factors with the $\delta$15N of leaf and soil is very ambiguous and unexplained. I would suggest to authors that the environmental factors and the response variables should be tested with principal component analysis(es) to get a clearer picture. The location of the two observatories on shady slope covers almost the whole range of the sampling gradient. However, on the sunny slope the two observatories merely cover half of the total gradient of the altitude sampled. How would the authors justify the use of climate data obtained from these observatories for the entire gradient of the altitude sampled and studied? Few general comments:

L 48: localized is a better word that 'local-dependent'. L 303: Various instead of varied. L 316-320: Should the plant not discriminate against the heavier isotope during N uptake, even if it's very low, thereby resulting in low leaf 15N signature, when higher N uptake is the routine? L 336-340: This explanation presented here just says that cold temperature caused high leaf $\delta$15N on shady slope. But how?

―――――――――――――――

---

## Referee Comment (RC2) · Anonymous Referee #2 · 8 Oct 2017

The authors report results from a study comparing the nitrogen isotopic values of plants and soils of north and south-facing slopes of an Asian mountain range. [At least it seems that they are north and south facing slopes, since the authors only talk about shady and sunny] At each site, they measured the delta15N of plants and soils as well as a series of other climatological and soil variables. These were used in a series of correlations and predictive models.

The authors report that "sunny" slopes have higher leaf and soil delta15N. They also report different factors affect delta15N on sunny and shady slopes.

The paper would be clearer if the authors referred to north- and south-facing slopes,

not sunny and shady. If this is wrong, the authors need to describe how a slope was determined to be either sunny or shady.

The authors interpolate mean annual temperature and mean annual precipitation for each site from measurements of MAT and MAP from 4 climate stations, two of which are sunny and two of which are shady. This is not valid. The authors sites are varying by a number of factors that cannot be "interpolated" from just 4 points. Stating that sunny sites are warmer than shady sites will need other data. One recommendation would be to simply remove the MAT and MAP regressions/correlations and examine other factors.

The authors interpret the difference of leaf and soil delta 15N as "as the isotopic composition of plant-available N". There is no empirical evidence for this. Given the results of Craine et al. 2015 that examines global patterns of soil 15N, there is unlikely to be evidence that the signature of available N is controlled by soil delta 15N. soil delta 15N at broad scales is likely simply an index of decomposition of the soil organic matter. Unless the authors have a reference to a graph that shows directly this relationship (delta 15N of available N vs. soil delta 15N) this statement is poorly supported.

Figure 1 needs to redrawn at a much larger scale, i.e over less total area. The points all overlap and it is not helpful to see where the sampling is.

---

## Author Response (AR1)

Dear Dr. Sébastien Fontaine,

Thank you and the reviewers very much for the kind and irradiative comments on our manuscript entitled "*Nitrogen isotopic composition of plants and soil in an arid mountainous terrain: sunny slope versus shady slope*" (The revised title is "Nitrogen isotopic composition of plants and soil in an arid mountainous terrain: south slope versus north slope", bg-2017-313). These comments are valuable and very helpful for the improvement of our manuscript.

We have revised our manuscript after carefully reading the comments. The revised parts were marked in **red color** in the "marked-up manuscript version". The detailed replies or explanations on the comments were given on the following pages.

We wish this revised manuscript meet the demands for publication in "*Biogeosicences*". Thank you again!

Best regards

Yours sincerely,

Guoan Wang

E-mail address: gawang@cau.edu.cn

China Agricultural University

**Answers to the questions:**
* * *
**Reviewer #1:**

**1. Response to comment 1,** The reasoning behind different results and the varying environmental factors determining them on two slopes is not clear. In addition, the correlation of various environmental factors with the $\delta^{15}$N of leaf and soil is very ambiguous and unexplained. I would suggest to authors that the environmental factors and the response variables should be tested with principal component analysis(es) to get a clearer picture.

**Answer:** Special thanks to you for your good comment. According to your advice, we tested all variables using principal component analysis, the results was displayed in the following figure. In principal component analyses, PC1 and PC2 could represent soil conditions and plant traits (especially leaf N content), respectively. The results of principal component analyses (Fig. 4) seem consistent with correlation analyses (please see the following Tables 3 and 4). On the north slope, leaf N content had strong positive while leaf C/N had negative effects on leaf $\delta^{15}$N, MAT and MAP also exerted influences on leaf $\delta^{15}$N, however, soil factors almost did not affect leaf $\delta^{15}$N except silt/clay ratio and soil moisture. Both MAT and MAP had large loadings on soil $\delta^{15}$N, meanwhile, soil $\delta^{15}$N increased with decreasing silt/clay ratio and increasing soil moisture. Compared with the north slope, representation of PC1 and PC2 on the south slope was clearer. Leaf $\delta^{15}$N was primarily correlated with leaf C/N, soil $\delta^{15}$N was significantly controlled by MAP and soil moisture, which might be due to arid environment on the south slope. Principal component analyses and correlation analyses both supported our argument that the relationships between leaf and soil $\delta^{15}$N and environmental factors are localized (lines 216-218, 245-248, 250-252, 276-278, 616-621).

[Figure]

**Fig 4. Variables loading on the first two principle components of the north (a) and south slope (b).**

**Table 3. Correlation analyses between leaf or soil $\delta^{15}N$ and influential factors on the north slope of Mount Tianshan.**

| | Leaf $\delta^{15}N$ | | Soil $\delta^{15}N$ | |
|---|---|---|---|---|
| | *r* | *P* | *r* | *P* |
| Leaf $\delta^{15}N$ | 1 | --- | -0.120 | 0.264 |
| Soil $\delta^{15}N$ | -0.120 | 0.264 | 1 | --- |
| MAT | **0.266** | 0.012 | **-0.385** | < 0.001 |
| MAP | **-0.272** | 0.010 | **0.387** | < 0.001 |
| Leaf N content | **0.340** | 0.001 | -0.090 | 0.397 |
| Leaf C/N | **-0.452** | < 0.001 | -0.036 | 0.739 |
| Soil N content | -0.048 | 0.659 | 0.088 | 0.408 |
| Soil moisture | **-0.271** | 0.011 | **0.388** | 0.000 |
| Soil pH | 0.162 | 0.132 | 0.070 | 0.513 |
| Soil bulk density | -0.056 | 0.604 | 0.145 | 0.174 |
| Silt/clay ratio | **-0.236** | 0.027 | **-0.370** | < 0.001 |

*Note: the r values were in bold when P < 0.05.*

**Table 4. Correlation analyses between leaf or soil $\delta^{15}N$ and influential factors on the south slope of Mount Tianshan.**

| | Leaf $\delta^{15}N$ | | Soil $\delta^{15}N$ | |
|---|---|---|---|---|
| | *r* | *P* | *r* | *P* |
| Leaf $\delta^{15}N$ | 1 | --- | 0.175 | 0.074 |
| Soil $\delta^{15}N$ | 0.175 | 0.074 | 1 | --- |
| MAT | 0.157 | 0.109 | 0.115 | 0.244 |

| | | | | |
|---|---|---|---|---|
| MAP | -0.168 | 0.087 | **-0.203** | 0.038 |
| Leaf N content | 0.119 | 0.229 | -0.073 | 0.459 |
| Leaf C/N | **-0.228** | 0.021 | 0.062 | 0.533 |
| Soil N content | -0.173 | 0.078 | 0.014 | 0.888 |
| Soil moisture | -0.141 | 0.150 | **-0.229** | 0.019 |
| Soil pH | 0.04 | 0.686 | -0.138 | 0.161 |
| Soil bulk density | 0.151 | 0.125 | 0.041 | 0.679 |
| Silt/clay ratio | -0.07 | 0.477 | -0.004 | 0.964 |

*Note: the r values were in bold when P < 0.05.*

**2. Response to comment 2,** The location of the two observatories on shady slope covers almost the whole range of the sampling gradient. However, on the sunny slope the two observatories merely cover half of the total gradient of the altitude sampled. How would the authors justify the use of climate data obtained from these observatories for the entire gradient of the altitude sampled and studied?

**Answer:** Your comment is right! In this paper, MAT and MAP were interpolated by two observations on each slope. We have to admit that the interpolated climatic data might be not very reliable, but we have no better ways to obtain more reliable climatic data. It is well known that this is also the greatest difficulty that the researchers studying global changes encounter. In fact, the case that two observations distributed at each slope is very rare in the world, and this is also one reason why we conducted the investigation here.

**3. Response to comment 3,** L48: localized is a better word that "local-dependent".

**Answer:** Thanks, "local-dependent" has been changed to "localized" in revised manuscript (lines 48, 107, 321).

**4. Response to comment 4,** Various instead of varied.

**Answer:** Thanks, "varied" has been changed to "various" in revised manuscript (lines 316).

**5. Response to comment 5,** L316-320: Should the plant not discriminate against the heavier isotope during N uptake, even if it's very low, thereby resulting in low leaf $^{15}$N signature, when higher N uptake is the routine?

**Answer:** Sorry. We did not offer a clear explanation in the original manuscript. We did changes for this in the new version. The explanation is as follows. This is a widely accepted fact that plants are depleted in $^{15}$N relative to its N sources because of $^{15}$N discrimination, but in this paper, we meant that the plants grown in N-limited environments will enrich more $^{14}$N compared with the plants in N-rich condition. The reason is that soil N transformations, such as $NH_3$ volatilization and $NO_x$ emission are enhanced when soil N nutrient is rich, consequently, more $^{14}$N losses from soil. This causes $^{15}$N enrichment in soil, subsequently, plant $\delta^{15}$N is more positive. Conversely, plants have more negative $\delta^{15}$N values when soil N is limited because of weak soil N transformations and less $^{14}$N loss (lines 331-339).

**6. Response to comment 6,** L336-340: This explanation presented here just says that cold temperature caused high leaf $\delta^{15}$N on shady slope. But how ?

**Answer:** Sorry. We did not present a detailed mechanism for this (a positive effect of temperature on the north slope) in the manuscript, and the reason is that we are not sure about the mechanism. The probable mechanism is that higher temperature favors more complete plant nitrogen assimilation and transformation, which might decrease isotopic fractionation during N assimilation and transformation, then causes $^{15}$N enrichment in plants. We will add the probable mechanism in the new version (lines 355-369, 511-512, 539-544).

**Reviewer #2:**

**1. Response to comment 1,** The paper would be clearer if the authors referred to north- and south-facing slopes, not sunny and shady. If this is wrong, the authors need to describe how a slope was determined to be either sunny or shady.

**Answer:** According to the reviewer's comment, we have changed shady and sunny slope to north and south slope throughly in revised manuscript.

**2. Response to comment 2,** The authors interpolate mean annual temperature and mean annual precipitation for each site from measurements of MAT and MAP from 4 climate stations, two of which are sunny and two of which are shady. This is not valid. The authors sites are varying by a number of factors that cannot be "interpolated" from just 4 points. Stating that sunny sites are warmer than shady sites will need other data. One recommendation would be to simply remove the MAT and MAP regressions/correlations and examine other factors.

**Answer:** In this paper, MAT and MAP were interpolated by two observations on each slope. We have to admit that the interpolated climatic data might be not very reliable, but we have no better ways to obtain more reliable climatic data. In fact, the case that two observations distributed at each slope is very rare in the world. Although the obtained climatic data were not very reliable, we think it is necessary to remain the regressions/correlations between $\delta^{15}$N and MAT and MAP in this paper. The reason is that, MAT and MAP effects on leaf and soil $\delta^{15}$N at global scale are different from that at regional scale, this led to our argument (hypothesis) that environmental effects on leaf and soil $\delta^{15}$N could depend on local environment, thus, a comparative study between the south and north slope with significant differently climate conditions was conducted. The regressions/correlations between $\delta^{15}$N and MAT and MAP in this study did confirm our argument. If the MAT and MAP regressions/correlations were removed, our argument (hypothesis) will loss supports. Lacking reliable climatic data is a universal and most trouble for the researchers studying global change and biogeochemistry cycles. Although the regressions/correlations between $\delta^{15}$N and MAT and MAP obtained in this study could be not perfect or reliable due to lacking accurate climatic data, we believe that the present study is also a small progress in science because we first put forward this argument (hypothesis), and confirm it. We hope more researchers to follow it.

Besides climatic data, vegetation types and species provide a strong support for the warmer climate on the south slope than the north slope. The main species occurred on the south slope were *Ephedra sinica*, *Stipa grandis*, *Stipa capillata*, *Achnatherum splendens*, *Nitraria tangutorum*, *Caragana sinica*, and *Suaeda glauca*, all these plants are typical xerophyte species, and they were not found on the north slope. On the north slope, the main species included *Kobresia myosuroides*, *Carex enervis, Poa annua* and *Thalictrum aquilegifolium*, they all are not xerophyte species. The information was added in the revised version (lines 139-140, 147-149).

**3. Response to comment 3,** The authors interpret the difference of leaf and soil delta 15N as "as the isotopic composition of plant-available N". There is no empirical evidence for this. Given the results of Craine et al. 2015 that examines global patterns of soil 15N, there is unlikely to be evidence that the signature of available N is controlled by soil delta 15N. Soil delta 15N at broad scales is likely simply an index of decomposition of the soil organic matter. Unless the authors have a reference to a graph that shows directly this relationship (delta 15N of available N vs. soil delta 15N) this statement is poorly supported.

**Answer:** Amundson et al. (2003) suggested that $\Delta\delta^{15}$N$_{\text{leaf-soil}}$ could be interpreted as the isotopic composition of plant-available N provided that isotopic discrimination does not occur during plant uptake and assimilation. In the present study, we found a highly correlation between leaf $\delta^{15}N$ and $\Delta\delta^{15}N_{\text{leaf-soil}}$ both on the two slopes, which is consistent with the result in Craine et al. (2009). As we all recognized, leaf $\delta^{15}N$ is a good indicator of plant N sources characteristics. Thus, the relationship between leaf $\delta^{15}N$ and $\Delta\delta^{15}N_{\text{leaf-soil}}$ could provide a powerful support for the suggestion in Amundson et al. (2003).

Besides, even though there was no direct evidence to support the relationship between $\delta^{15}N$ of bulk soil N and $\delta^{15}N$ of available N, the statement that soil $\delta^{15}N$ could be used to index the soil N availability had been widely accepted (McLauchlan et al., 2007; Högberg, 1997). The mechanism was that, high soil N availability leads to increased soil N transformation, such as nitrification, denitrification and $NH_3$ volatilization, which discriminates against $^{15}N$ and causes $^{15}N$-enrichment in soil. Thus, ecosystems with high N availability exhibit high $\delta^{15}N$ values in soil (lines 303-314).

**4. Response to comment 4,** Figure 1 needs to redraw at a much larger scale, i.e over less total area. The points all overlap and it is not helpful to see where the sampling is.

**Answer:** Thank you for your advice, we have redrawn Figure 1 in revised manuscript (lines 568-575).

[revised manuscript text omitted]